# Focus on the Most Common Paucisymptomatic Vasculopathic Population, from Diagnosis to Secondary Prevention of Complications

**DOI:** 10.3390/diagnostics13142356

**Published:** 2023-07-13

**Authors:** Eugenio Martelli, Iolanda Enea, Matilde Zamboni, Massimo Federici, Umberto M. Bracale, Giuseppe Sangiorgi, Allegra R. Martelli, Teresa Messina, Alberto M. Settembrini

**Affiliations:** 1Department of General and Specialist Surgery, Faculty of Pharmacy and Medicine, Sapienza University of Rome, 155 Viale del Policlinico, 00161 Rome, Italy; 2Medicine and Surgery School of Medicine, Saint Camillus International University of Health Sciences, 8 Via di Sant’Alessandro, 00131 Rome, Italy; 3Division of Vascular Surgery, Department of Cardiovascular Sciences, S. Anna and S. Sebastiano Hospital, Via F. Palasciano, 81100 Caserta, Italy; 4Emergency Department, S. Anna and S. Sebastiano Hospital, Via F. Palasciano, 81100 Caserta, Italy; i_enea@hotmail.com; 5Division of Vascular Surgery, Saint Martin Hospital, 22 Viale Europa, 32100 Belluno, Italy; wambazamba@icloud.com; 6Department of Systems Medicine, School of Medicine and Surgery, University of Rome Tor Vergata, 1 Viale Montpellier, 00133 Rome, Italy; federicm@uniroma2.it; 7Division of Vascular Surgery, Federico II Polyclinic, Department of Public Health, School of Medicine and Surgery, University of Naples Federico II, 5 Via S. Pansini, 80131 Naples, Italy; umbertomarcello.bracale@unina.it; 8Department of Biomedicine and Prevention, School of Medicine and Surgery, University of Rome Tor Vergata, 1 Viale Montpellier, 00133 Rome, Italy; gsangiorgi@gmail.com; 9Faculty—Medicine & Surgery, Campus Bio-Medico University of Rome, 21 Via À. del Portillo, 00128 Rome, Italy; allegramartelli02@gmail.com; 10Division of Anesthesia and Intensive Care of Organ Transplants, Umberto I Polyclinic University Hospital, 155 Viale del Policlinico, 00161 Rome, Italy; teremessina@hotmail.com; 11Division of Vascular Surgery, Maggiore Polyclinic Hospital Ca’ Granda IRCCS and Foundation, 35 Via Francesco Sforza, 20122 Milan, Italy; amsettembrini@gmail.com

**Keywords:** diagnosis, secondary prevention, complications, abdominal aortic aneurysm, carotid stenosis, lower extremity arterial disease, popliteal artery aneurysm, renal artery stenosis, subclavian artery stenosis

## Abstract

Middle-aged adults can start to be affected by some arterial diseases (ADs), such as abdominal aortic or popliteal artery aneurysms, lower extremity arterial disease, internal carotid, or renal artery or subclavian artery stenosis. These vasculopathies are often asymptomatic or paucisymptomatic before manifesting themselves with dramatic complications. Therefore, early detection of ADs is fundamental to reduce the risk of major adverse cardiovascular and limb events. Furthermore, ADs carry a high correlation with silent coronary artery disease (CAD). This study focuses on the most common ADs, in the attempt to summarize some key points which should selectively drive screening. Since the human and economic possibilities to instrumentally screen wide populations is not evident, deep knowledge of semeiotics and careful anamnesis must play a central role in our daily activity as physicians. The presence of some risk factors for atherosclerosis, or an already known history of CAD, can raise the clinical suspicion of ADs after a careful clinical history and a deep physical examination. The clinical suspicion must then be confirmed by a first-level ultrasound investigation and, if so, adequate treatments can be adopted to prevent dreadful complications.

## 1. Introduction

This state-of-the-art article focuses on screening for the most common arterial diseases (ADs) in the over 50–60-year-old population to prevent dreadful complications using medical, endovascular, or surgical therapy (that is, secondary prevention). ADs start as asymptomatic or paucisymptomatic, and should be searched for when middle-age patients with one or more risk factors for atherosclerosis seek medical consultation, for instance to a cardiologist or a pneumoligist. Therefore, for each ADs described in this article, we aim to answer two questions: (1) what and to whom screening is for, and (2) how to prevent cardiovascular complications. This content is aimed toward the medical community, i.e., general practitioners, internists, nephrologists, geriatricians, cardiologists, but also to surgeons.

## 2. Relevant Sections

### 2.1. Abdominal Aortic Aneurysm (AAA)

An artery is defined as an aneurysm if its maximal transverse diameter exceeds the average diameter by at least 50%. Around 80% of the aneurysms of the aorta, exclusively or in part, involve the abdominal portion: compared to thoracic aorta aneurysms, AAAs are easier to diagnose (i.e., by the means of physical examination and duplex scan investigation). Almost 90% of AAAs are localized below the renal arteries (Figure 1).

The diameter of the infrarenal aorta in normal adults is in the range of 1.41–2.39 cm in men and 1.19–2.16 cm in women [1]. Therefore, we should refer to the diameter of the aorta immediately above the dilation, before eventually declaring that that dilation is effectively an AAA.

AAAs are almost always asymptomatic. Sometimes the patient reports feeling “the heart in the belly” when relaxing in a bed or chair. Endovascular or open treatment is indicated when the maximal transverse diameter reaches or exceeds 5.5 cm. At this dimension, the annual rupture rate (and the consequent fatal hemorrhage) is 5–10% per year, which is higher than the estimated rate of elective operative complications (around 2–5% in dedicated vascular centers). The rupture rate increases proportionally with the growth of the AAA, and a ruptured AAA (rAAA) rarely gives a warning [2].

AAA is more common in males but if present in females presents a higher rupture risk. It often manifests with abdominal or back pain and hypotension: too late to avoid an overall mortality rate that can be up to 80% [3,4]. This is why detecting an asymptomatic intact AAA practically means saving a patient’s life.

(1)Who to screen and how to screen for AAA?

The etiopathogenesis of AAA is considered multi-factorial, involving inflammation, proteolysis, vascular smooth muscle apoptosis, and the following risk factors: male sex, older age, tobacco use, family history, hypertension, hypercholesterolemia, and history of other aneurysms [5,6,7,8]. Therefore, patients displaying these risk factors should be particularly focused on for AAA screening.

A meta-analysis of population-based randomized control trials (RCTs) established that screening of men ≥65 years of age was associated with both increased rates of AAA detection and elective repair, and both lower AAA-related mortality and rAAA incidence for up to 15 years [9]. Similar results were shown in a recent scoping review, which also reported the cost effectiveness of reducing emergency rAAA treatment, but also outlined a few potential harms of older male AAA screening: no AAA mortality or morbidity screening-related benefits, a negative impact on men’s quality of life, inconsistent application of AAA screening recommendations by primary care practitioners, and a tendency to repair AAAs smaller than the recommended threshold [10,11].

Since current studies have demonstrated an overall decline in AAA prevalence, there is a need for a customized AAA screening age range for older adult men, who should be evaluated individually by taking risk factors and estimated life expectancy into account [12,13,14]. The family history of AAA should be better investigated to address the correct screening. A study from Sweden and Denmark pointed out a 24% incidence in the monozygotic twin of an AAA patient; incidence seemed to increase in the case of women; these AAAs grew up faster and could rupture even if smaller than 5 cm [15]. Hopefully, the results of an ongoing Swedish study will answer some open questions, such as screening for AAA in older siblings, women included.

Most of the studies on this topic investigated only more senior men, likely because the proportion of women with an intact AAA is low compared to men (1,4–6 ratio); still, their rate of rupture is higher [16,17]. Some Danish colleagues have recently underlined the need for better screening for AAA in women. They noticed poor agreements in duplex scan-based absolute diameters concerning aortic ectasias, small AAAs, and large AAAs in women, indicating that the current absolute cut-points do not reflect female anatomy [18].

Colleagues from the University of Pittsburgh (Pittsburgh, PA, USA) demonstrated that among 632 patients who had undergone treatment for rAAA over a 17-year period, residing in the most deprived neighborhoods was associated with a greater probability of presenting rAAA under age 65 and, therefore, may benefit from a younger screening age [19].

A Swiss study showed that among 213 consecutive rAAAs emergently operated on between 1998 and 2005, all patients had been seen by their general practitioner or a cardiologist within a year before the presentation. Therefore, they may have benefited from AAA screening [20].

Regarding aortic ectasia (i.e., with an absolute diameter between 2.5–2.9 cm), due to cost containment, Sweden is the only country that recommends follow-up surveillance at five years [21]. About one-quarter of aortic ectasias can develop into AAA greater than 5.5 cm at 10 years, and eventually rupture [22,23].

Sometimes, an intact asymptomatic AAA is accidentally detected during instrumental investigations conducted in the context of other disease: for instance, a duplex scan of the liver or of the prostate, or computed tomography (CT-scan) or magnetic resonance imaging (MRI) of the column.

Establishing an efficient and cost-effective screening methodology for asymptomatic AAAs is paramount, since it allows for early detection, surveillance, and intervention to prevent rAAA. During a routine physical examination, the deep palpation of the abdomen is a fundamental principle in searching for an abnormal pulsating mass in the mesogastrium (Figure 2).

Then, the eventual clinical suspect of an AAA needs to be confirmed or excluded by a first-level investigation, such as a duplex scan, which has a very high specificity (almost 100%) and sensitivity (95%) to detect an AAA (Figure 3) [24].

These concepts should be routinely adopted by general practitioners, internists, and cardiologists. These are often the only referral physicians for many patients. Aortic ultrasound investigations are usually performed by radiologists, but also by vascular physicians or vascular surgeons. If an AAA is diagnosed, the patient must be referred to a vascular specialist for the most appropriate management (Figure 4).

(2)How to prevent rAAAs?

AAAs ≥ 5–5.5 cm undergo evaluation for operative treatment. Preoperative CT-scan of the abdomen with contrast medium is mandatory in case of endovascular exclusion of the AAA, and is strongly suggested in case of open repair. On the other hand, AAA < 5–5.5 cm enter a follow-up program by duplex scan. Aneurysm diameter remains a well-established parameter for clinical decision making.

Chronic obstructive pulmonary disease and active smoking are risk factors associated with AAA expansion [25]. The AAA annual growth rate is increased in smokers (by 0.35 mm/year), decreased in patients with diabetes (by 0.51 mm/year), is correlated to the diameter (around 1.3 mm for 3 cm AAA, 3.6 mm for those of 5 cm), and does not significantly differ between the genders. The risk of AAA rupture is increased substantially in women, older patients, and in those with lower body mass index (BMI) or higher blood pressure (BP) [26].

### 2.2. Extracranial Internal Carotid Stenosis (EICS)

Around 85% of strokes are of ischemic origin. Atrial fibrillation is a major cause of cerebral embolization, causing more transient ischemic attacks (TIAs) than strokes, but 15–20% of all strokes are due to atherosclerotic obstructive lesions of the EICS [27,28]. Therefore, treating EICS due to an atherosclerotic plaque means secondary prevention of cerebral ischemia.

The pathogenetic mechanism is mainly embolic or related to cerebral hypoperfusion in case of severe occlusive disease in both the carotid and vertebral arteries.

(1)For whom and how to screen for EICS?

A meta-analysis of four population-based studies reported a 3.1% prevalence of severe asymptomatic EICS in the general population. A large USA and UK population of 2.5 million people screened for EICS showed that the prevalence of EICS is between 2% and 2.3% for ages 60–69, 3.6% and 6% for ages 70–79, and 5% and 7.5% for those >80 years old, for women and men, respectively. The probability of detecting an EICS is three times higher in smokers, more than two times higher if systolic BP > 160 mmHg, and in diabetics, 40% higher for each one mmol/L increase in low-density lipoprotein (LDL) cholesterol, 40% lower for each one mmol/L increase in high-density lipoprotein (HDL) cholesterol, 75% higher for the doubling of triglycerides, and 20% more elevated for each 5-unit increase in body mass index >25 kg/m^2^ [29].

Therefore, once again, an accurate clinical history in middle-aged adults displaying these risk factors is mandatory. In addition to risk factors, these patients should be investigated for any sudden, transient symptom suggestive of carotid pathogenesis, such as paresis or paresthesias in one or both ipsilateral limbs, aphasia, and amaurosis fugax. Quite often, the anamnesis is confounded by symptoms of vertebrobasilar insufficiency, such as vertigo, ataxia, visual disturbances, and diplopia.

Thus, the search for carotid bruits during physical examination is paramount. The vascular bruit is the acoustic manifestation of the turbulence caused by blood flow passing through an arterial stenosis. Therefore, a bruit over the carotid pulse could imply the presence of a stenosis on the internal, external, or common carotid artery, hence requiring a duplex scan investigation. Sometimes, the absence of a carotid bruit represents false negative data, for instance, when an eventual carotid stenosis is so tight that turbulence in the blood flow is not even generated. Other times, bilateral carotid and subclavian artery bruits represent false positive data, very likely being bruits transmitted from an aortic valve stenosis. In any case, this simple semeiologic maneuver should always be kept in mind during routine physical examination in middle-aged patients with risk factors for atherosclerosis.

Regardless, patients older than 65 with two or more risk factors for atherosclerosis, coronary artery disease (CAD), or ADs in other districts are at high cardiocerebrovascular risk, and should be screened with a duplex scan in the search for EISC [30]. General practitioners, internists, cardiologists, neurologists, and ophthalmologists are most often involved with these patients, and should send them to radiologists or vascular physicians for a carotid duplex scan (Figure 5).

(2)How to prevent an ischemic stroke of EICS origin?

What makes the difference in stroke prevention in a patient with EICS is if the patient is asymptomatic or transiently symptomatic for congruous hemispheric or retinal ischemia in the last three months [31]. It is well established that transiently symptomatic carotid stenosis is best managed with intervention, either by carotid endarterectomy (CEA) or stenting. In the short term, TIA precedes 15% of ischemic strokes [32]. Independent risk factors are an age greater than 60 years, diabetes mellitus (DM), focal symptoms, and TIAs that last longer than 10 min [33]. Chronic renal insufficiency (CRI) is another risk factor for stroke in patients with asymptomatic EICS [34].

On the other hand, the optimal management of asymptomatic carotid stenosis is still controversial. An asymptomatic >50% (according to the North American Symptomatic Carotid Endarterectomy Trial (NASCET) criteria) EICS carries a 1% annual risk of stroke. Most of the merits of this low incidence are due to the development of the best medical therapy (BMT), comprising smoking cessation, antiplatelet agent use, cholesterol reduction, exercise therapy, and the diagnosis and treatment of high BP and DM. Every 10% decrease in LDL cholesterol reduces the risk of stroke by 15% in patients with significant EICS [35]. Low-dose acetylsalicylic acid (81 mg qd) is the most commonly used cardiovascular prophylaxis. Every 10 mmHg increase in BP worsens the risk of stroke by 30–45% [36,37]. Furthermore, smoking increases the risk of stroke by 1.5% [38]. Therefore, in the last 10 years, BMT has been increasingly considered the treatment of choice for many asymptomatic EICSs [39,40].

The risk of stroke increases according to the degree of EICS: those greater than 80% carry an annual risk of 4.8% [29]. Duplex scan is a first-level investigation to determine the degree of EICS utilizing peak systolic velocity (Figure 6).

In complex duplex visualization, as in the presence of a highly calcific carotid plaque causing an ultrasound shadow cone, the exact degree of EICS can be accurately measured by computed tomographic angiography (CTA), or eventually by digital subtraction angiography (DSA). Compared to the latter, CTA tends to underestimate the higher and moderate grade EICS [41]. Magnetic resonance angiography (MRA) represents another second-level instrumental examination in patients contraindicated to iodine-based contrast. MRA underestimates EICS since it is not as sensitive to calcification. On the other hand, gadolinium contrast-enhanced MRA can overestimate EICS since it is more impacted by artifacts [42]. Either the NASCET criteria or European Carotid Surgery Trialists (ECST) criteria are used to calculate EICS (Figure 7) [43].

In the NASCET criteria, severe and moderate EICS is defined as 70–99% and 50–69% stenosis, respectively. In ECST criteria, severe and mild EICS is defined as 80–99% and 70–79% stenosis, respectively [44].

Together with the degree of EICS, another essential predictor of cerebral ischemia is identifying a vulnerable plaque. Intraplaque hemorrhage, discontinuous fibrous cap, or frankly, ulceration, echolucency at the duplex scan, mural thrombus, neovascularization and inflammation, lipid-rich necrotic core, and microembolic signals at the transcranial doppler can be associated with thromboembolism from the lipid core, and represent an indication to open or endovascular treatment [45]. These plaque characteristics are found in 43.3% of symptomatic patients with EICS and only in 19.9% of the asymptomatic EICS [46]. Detection of active inflammation in the lipid core of a carotid plaque is an indicator of unstable plaque and rupture-prone fibrous cap. Clinical observations suggest that the calcification of the lipid core is stable, while the less-calcified atheroma is more prone to rupture. Therefore, chronic inflammation in a noncalcified plaque is crucial for atherosclerotic plaque vulnerability and disruption.

### 2.3. Lower Extremity Arterial Disease (LEAD)

LEAD is a progressive atherosclerotic occlusion of the arteries of the lower limbs. If not medically treated, according to the natural history of atherosclerosis, LEAD can chronically progress to chronic limb-threatening ischemia (CLTI), a condition characterized by rest pain and gangrene in the foot. Furthermore, according to Virchow’s triad, the endothelial lesions represented by the atherosclerotic plaques can be the sites of overimposed acute arterial thrombosis, so that LEAD can acutely complicate with acute limb ischemia (ALI). Both CLTI and ALI carry a high risk of major adverse cardiovascular events (MACE), that is, acute myocardial infarction and stroke, and major adverse limb events (MALE), up to major amputation [47,48].

Furthermore, LEAD itself is an independent predictor of MACE. Its incidence increases significantly with age, showing a 20% prevalence peak in the over-80 population [49].

Fortunately, if compared to the ADs dealt with before, LEAD is more symptomatic, hence facilitating its diagnosis. However, only one-third of LEAD patients present the typical intermittent claudication (IC), a cramping pain in some muscles of the lower limbs, which arises while climbing stairs or walking and recedes within a few minutes of resting [50].

(1)Who and how to screen for LEAD?

There are two categories of patients possibly displaying LEAD:-Those who have a masked LEAD, since they do not present IC: reasons range from the development of efficient arterial collateral circulation (Figure 8) to limited mobility, up to being bedridden [51];-The 30–50% of patients who have an atypical symptomatology due to the co-existence of other diseases, such as disc herniation, medullary stenosis, vertebral osteoarthritis, and peripheral neuropathy [52].

These asymptomatic LEAD patients should also be identified to optimize atherosclerotic risk factor control and medical therapy to reduce their silent risk of cardiovascular mortality. LEAD must be suspected in adults with DM, which carries a four-fold risk compared to the general population [53]; cigarette smoking, a two-fold risk [54]; high BP, the strongest predictor of LEAD [55]; hyperlipemia [56]; familiarity [57]; and hyperfibrinogenemia [58]. Furthermore, the prevalence of LEAD is higher in patients with chronic renal function impairment, and increases moving from mild to severe degrees [59].

An exhaustive clinical history should search for symptoms of IC, and make a differential diagnosis with other pathogenesis of pain in the lower limbs. Then, an accurate physical examination should bilaterally evaluate the presence or absence of the femoral, popliteal, anterior tibial and posterior tibial peripheral pulses, and femoral bruit. In the large majority of cases, the absence of a peripheral pulse means that the artery immediately above is almost or totally obstructed, that is the patient is affected by severe atherosclerosis. The presence of a femoral bruit is a sign of hemodynamic arterial stenosis in the aorto-iliac district.

When a general practitioner, internist, cardiologist, nephrologist, or geriatrician has the suspect of LEAD, instrumental examinations such as ankle-brachial index (ABI) or arterial duplex scan of the lower limbs need to be performed to confirm or exclude it (Figure 9).

The ankle-brachial index (ABI) is the easiest way to confirm the clinical suspect of LEAD or its screening. It is the ratio between two systolic BPs registered utilizing continuous wave Doppler in the supine patient, the one on an artery at the ankle, and the systemic one. A typical value is around 1–1.1, even up to 1.3, and LEAD is diagnosed if ABI is less than 0.9 (Figure 10).

The reduction in ABI’s value is directly proportional to the stage of the atherosclerotic disease, the risk of MACE, MALE, and walking impairment [46]. ABI can result in being paradoxically higher than 1.3 in subjects affected by DM or uremia due to the stiffness of their tibial arteries. The toe-brachial index (TBI) gives these patients more realistic values. TBI less than 0.7 is diagnostic for LEAD [60]. Sometimes ABI can be normal, even if the suspicion of LEAD is high. In these cases, a treadmill test can be helpful: if the following ABI shows a 20% reduction compared to the value at rest, LEAD is confirmed [61].

(2)How to prevent MACE and MALE (CLTI, ALI, and amputation)?

If LEAD is confirmed, the prevention of MACE and MALE should be accomplished utilizing a multidisciplinary team-based approach comprising cardiologists, endocrinologists, nephrologists, vascular medicine physicians, and vascular surgeons. The latter are the only ones who can offer a broad portfolio of therapeutic options in symptomatic LEAD patients, from medical to surgical treatment, from endovascular to open access, or hybrid revascularization.

Glycated hemoglobin should be in the range of 6.5–7.5% [62]. Smoking cessation is associated with a significative MACE reduction and improved functional outcomes and MALE in patients with IC. Systemic BP should be stabilized at 130/80 mmHg. The LDL cholesterol level inversely correlates with ABI, and should be kept to less than 55 mg/dL in patients at high cardiovascular risk. Intensive statin therapy, such as atorvastatin (40–80 mg) or rosuvastatin (20–40 mg qd), is recommended in these patients to reduce the risk of MACE and MALE and to improve the walking distance in the LEAD patients with IC. Antiplatelet therapy with low-dose (100 mg qd) acetylsalicylic acid, or ideally clopidogrel 75 mg qd, is strongly recommended to reduce the risk of ischemic events [51,52,63].

Vitamin K antagonist (warfarin) can reduce MACE, but the related risk of bleeding is not easy to be managed. Instead, the combined therapy of low-dose (2.5 mg bid) rivaroxaban (one of the four oral anticoagulants non-vitamin K antagonist) plus acetylsalicylic acid (100 mg qd) has been shown to reduce the risk of MACE and MALE compared to low-dose aspirin alone in symptomatic LEAD patients [64,65].

### 2.4. Popliteal Artery Aneurysm (PAA)

The diameter of the popliteal artery in normal adults is in the range of 0.7–1.1 cm in men and 0.5–0.9 cm in women [1]. Therefore, dilation of the popliteal artery is a PAA when its maximal transverse diameter exceeds the standard diameter of the popliteal artery immediately above the dilation by at least 50%.

PAAs should be detected not so much for the risk of rupture, which is rare, but for their risk of embolization and thrombosis. The physiologic flexion of the knee can act as a tremendous stress for the parietal thrombus of the PAA, potentially causing paroxysmal, multiple, and insidious episodes of asymptomatic microembolization to the tibial arteries (Figure 11). As a consequence, the latter can progressively obstruct, drastically reducing the run-off of the popliteal artery and giving rise to a clinical picture ranging from blue toe syndrome, or LEAD with IC up to chronic limb-threatening ischemia. PAA can also thrombose entirely due to the affected tibial out-flow, so it manifests itself with ALI: limb loss can reach 14% in these patients [66,67].

(1)Who and how to screen for PAA to?

PAA incidence in the general population can reach almost 3%, and men represent more than 90%. Almost 50% of the PAAs are bilateral, and over one-third are associated with AAA [66,68]. On the other hand, the incidence of a PAA in patients with an AAA is up to 11% [69].

Some patients with PAA may be clinically indistinguishable from those presenting with LEAD. Therefore, a careful examination is required from general practitioner, internist (nephrologist, geriatrician), and cardiologist to distinguish chronically symptomatic PAA patients from those with symptoms due to LEAD (Figure 12).

Physical examination searching for a pulsating mass in the popliteal lozenge is diagnostic for two-thirds of the PAAs. A typical differential diagnosis is with Baker’s synovial cyst, which can transmit the pulsation of a normal popliteal artery. Even in this case, duplex scan (performed by radiologists, vascular physicians, or vascular surgeons) has close to 100% diagnostic accuracy. Particular attention for screening must be given to patients with a personal or familial history of PAA (eventually contralateral), AAA, or aneurysms in other districts. Around 40% of PAAs are asymptomatic when detected, but up to 24% will become symptomatic in the next 1–2 years [70,71].

(2)How to prevent thromboembolism from PAA and rupture?

PAAs less than 2 cm in maximal transverse diameter can grow up to 1.5 mm annually [66]. As for the other ADs, these patients should cease smoking and control high BP, hyperlipidemia, and DM, as part of an atherosclerotic factor control strategy. PAA greater than 2 cm grows up to 3 mm annually and should be fixed in an open or endovascular fashion [72]. However, two more factors indicate surgical treatment: the presence and amount of the parietal thrombus and the poor distal run-off.

### 2.5. Renal Artery Stenosis (RAS)

RAS is associated with renovascular hypertension (RVH) and chronic renal insufficiency (CRI), so it configures the clinical picture of renovascular disease [73]. RAS greater than 50% causes dysregulation of the renin–angiotensin–aldosterone mechanism, which can cause RVH. The differential diagnosis between RVH and essential hypertension is difficult since specific symptoms and signs are lacking. However, RVH should be suspected in some patients since, if not adequately pharmacologically controlled, in the long term, it brings parenchymal alterations, even in the contralateral kidney, with reduced excretory capacity [74].

(1)Who and how to screen for RAS?

Most RAS is atherosclerotic, much less due to fibromuscular dysplasia or dissection. RAS prevalence, incidence, and natural history are largely unknown. An autoptic study reported 6% in under 55-year-old patients and 40% in those over the age of 75 [75]. At three years, RAS less than 60% showed progression to severe RAS in 40% of cases, and RAS greater than 60% showed progression to occlusion in 7% of patients [76].

The search for RAS passes through a high propensity for clinical suspicion in severely hypertensive patients resistant to medical therapy (RVH is more resistant to pharmacological treatment than essential hypertension) with CRI, advanced age, coronary artery disease, EICS, or LEAD (Figure 13).

A duplex scan is an excellent examination for RAS screening, even if its efficacy is limited in obese patients or with intestinal meteorism (Figure 14).

Thus, radiologists, vascular physicians, or vascular surgeons expert in renal artery duplex scan are required for RAS screening.

(2)How to prevent complications from RVH, end-stage renal failure, and kidney loss?

This complex cascade of events worsens the risk of MACE in patients with RAS [77]. It justifies the indication of renal artery surgical revascularization in highly selected patients with severe ostial RAS, short RAS occlusion, AAA associated with RAS, or aneurysm of the renal artery associated with RAS [78].

Renal artery stenting seems inferior to a well-conducted BMT [79]. The latter is aimed at reducing systemic BP, preserving the renal function, and preventing MACE, through substantial modifications of the patient’s lifestyle (correct BMI, no smoking, salt and alcohol reduction, regular physical activity) and specific medications (angiotensin receptor blockers, diuretics, β-blockers, antiplatelets, statins) [80].

### 2.6. Prevertebral Subclavian Artery Stenosis (PSAS)

A tight atherosclerotic stenosis (or occlusion) at the origin of the subclavian artery before the onset of its first collateral branch, the vertebral artery, can cause a brain stem blood steal during homolateral upper limb efforts, in favor of the latter, through a blood flow inversion in the homolateral vertebral artery itself. This condition is known as subclavian steal syndrome (SSS), which can acutely give rise to dramatic vertebrobasilar symptoms such as vertigo and lipothymia up to syncope (Figure 15).

(1)Who and how to screen for PSAS?

The prevalence of left PSAS is around 1.5% in the general population and over 11% in patients with LEAD. In comparison, right PSAS or brachiocephalic artery stenosis occurs less than one-quarter of the time, and the former is involved two-thirds of the time [81,82]. Given its peculiar hemodynamic pathophysiology, PSAS and SSS are mainly encountered in vasculopathics who are heavy workers or working with their upper limbs against gravity (ceiling restorers, orchestra directors, etc.).

Diagnosing PSAS is essential to prevent sudden and dangerous loss of consciousness due to SSS. General practitioners, internists, cardiologists, nephrologists, geriatricians, and also surgeons are called upon to raise awareness of the dangers arising from this condition. An unknown significative PSAS alters homolateral BP measurement, whose consequences can be dramatic during surgical procedures or in long-term therapy management in hypertensive patients. In a patient with coronary artery bypass graft (CABG) from the internal mammary artery (a branch of the subclavian artery) or with axillo-femoral bypass, an unknown significative homolateral PSAS can also cause ischemic events in the heart or in the lower limb, respectively. Furthermore, PSAS is an independent predictor of MACE [83].

In addition to the clinical history, physical examination is of utmost importance in the search for PSAS, since the clinical triad of differential BP > 15 mmHg between the upper limbs, subclavian bruit, and absence/hyposphygmia of the peripheral pulses in the homolateral upper limb is pathognomonic [84]. In particular, a differential BP of 30–20 mmHg between the upper limbs carries a high risk of SSS. Then, the clinical suspicion of PSAS can be confirmed utilizing a duplex scan, which has excellent diagnostic accuracy for the origin of the subclavian artery (Figure 16).

(2)How to prevent complications from PSAS?

Once detected, lifestyle measures and medical therapy common to other ADs are sufficient to manage asymptomatic PSAS. On the other hand, PSAS symptomatic for SSS or potentially placing a CABG from the homolateral internal mammary at risk, or an axillo-femoral bypass graft, should be revascularized. In this regard, angioplasty and stenting is the first-line treatment, eventually followed in case of endovascular unsuccess by carotid-subclavian bypass or subclavian transposition on the common carotid artery.

## 3. Conclusions

In addition to patients affected by CAD, the adult population over 50–60 years of age with one or more risk factors for atherosclerosis should be focused on for discovering asymptomatic or paucisymptomatic ADs, such as AAA, LEAD, EICS, PAA, RAS, and PSAS. Accurate anamnesis and deep physical examination can lead to the clinical suspect, which is confirmed or excluded by a duplex scan investigation in most cases. If even one of those ADs is diagnosed, that patient is to be considered affected by vasculopathy and should be managed by medical, endovascular, or surgical approaches to reduce the risk of MACE and MALE.

## 4. Future Directions

Since the world population is aging, the medical community is called upon to raise more awareness that ADs are increasing. EICS, LEAD, RAS, and PSAS correlate with the risk of MACE, and their detection can bring the discovery of silent CAD, reducing the risk of MACE itself. Since it is neither realistic or convenient to screen wide populations, accurate clinical history and deep physical examination remain key factors to select those individuals who should undergo first-level instrumental investigations to confirm or exclude the clinical suspect.

## Figures and Tables

**Figure 1 diagnostics-13-02356-f001:**
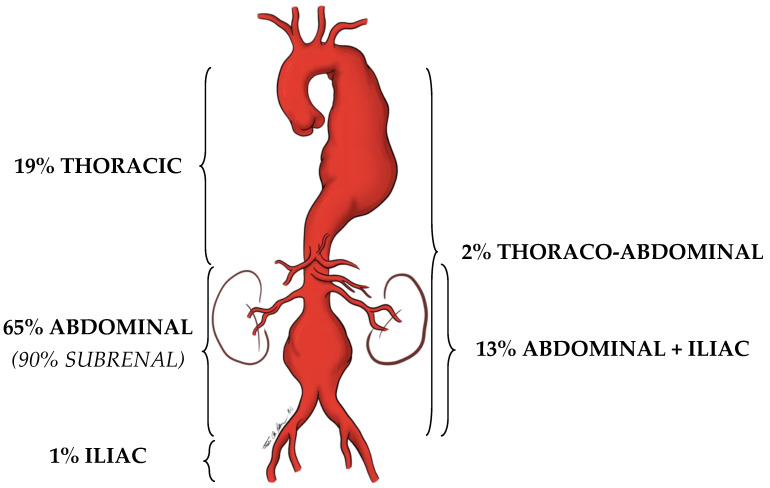
Localization prevalence of aortic aneurysms.

**Figure 2 diagnostics-13-02356-f002:**
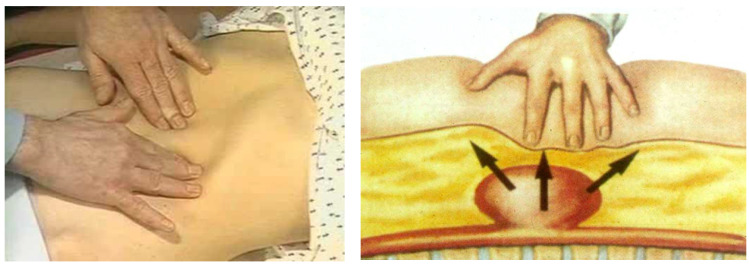
Abdomen palpation in the search for an abdominal aortic aneurysm (AAA). An abnormal pulsating mass in the mesogastrium, expanding in all the directions, is strongly suspected to be an AAA.

**Figure 3 diagnostics-13-02356-f003:**
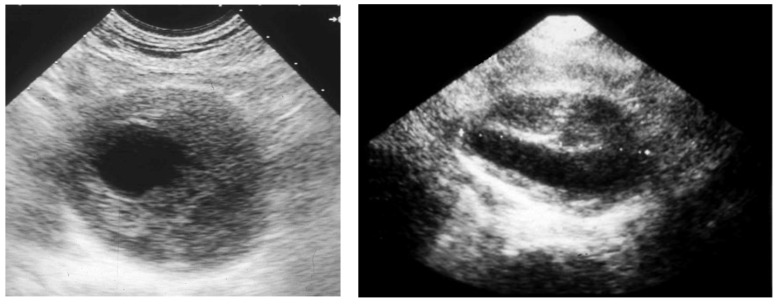
Duplex scan of the abdominal aorta can easily confirm the clinical suspect of abdominal aortic aneurysm. Short axis (**left**) and long axis (**right**) view.

**Figure 4 diagnostics-13-02356-f004:**
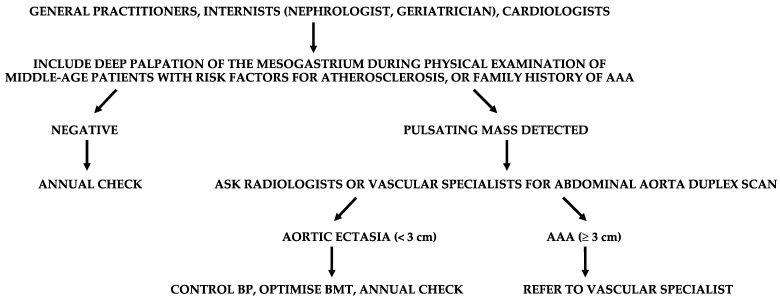
Suggested flow-chart for the detection of patients with abdominal aortic aneurysm (AAA). BP, blood pressure; BMT, best medical therapy.

**Figure 5 diagnostics-13-02356-f005:**
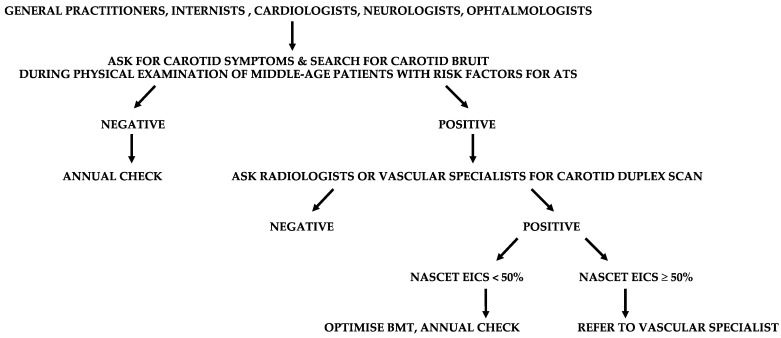
Suggested flow-chart for the detection of patients with extracranial internal carotid stenosis (EICS). ATS, atherosclerosis; NASCET (North American Symptomatic Carotid Endarterectomy Trial criteria); BMT, best medical therapy.

**Figure 6 diagnostics-13-02356-f006:**
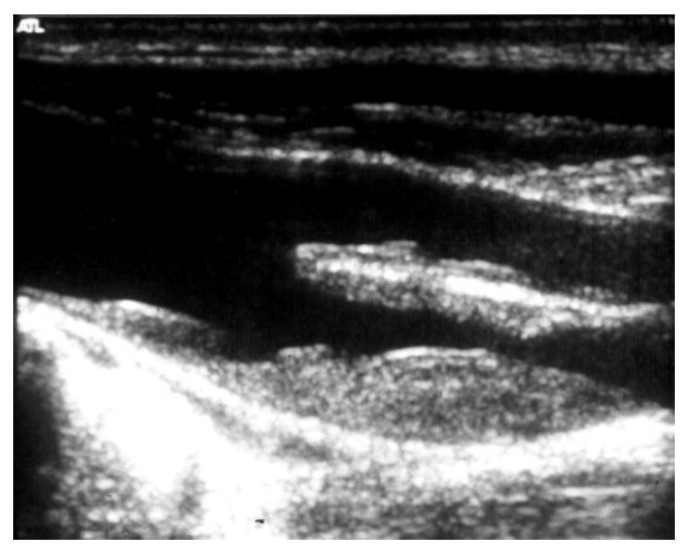
Duplex scan (long axis view) of the extracranial carotid bifurcation, clearly showing an atherosclerotic plaque causing stenosis of the internal carotid artery.

**Figure 7 diagnostics-13-02356-f007:**
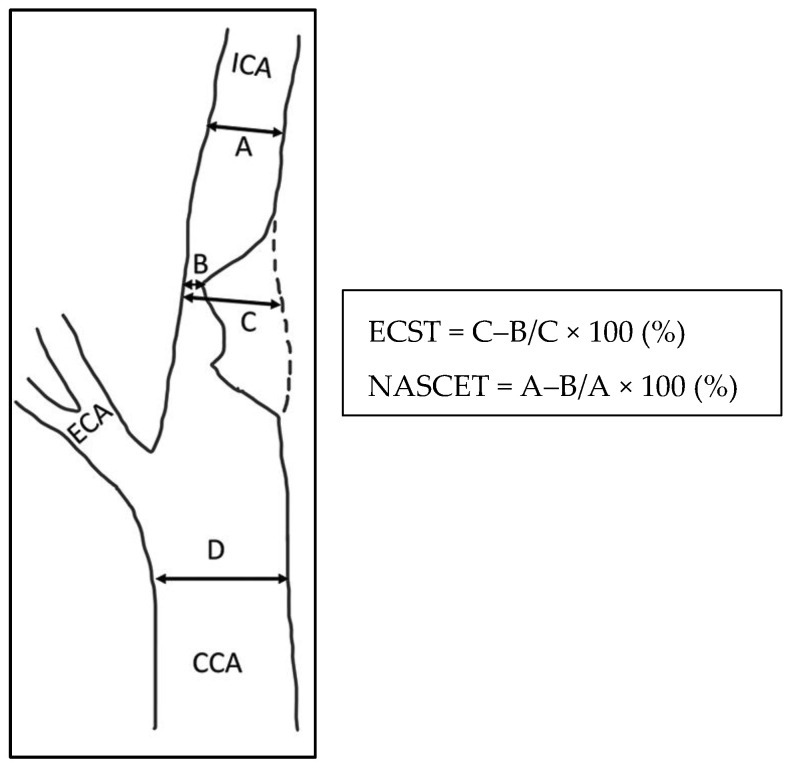
Methods of measurement of severity of ICA stenosis, reproduced with permission from Bir and Kelley [43]. CCA, common carotid artery; ICA, internal carotid artery; ECA, external carotid artery; ECST, European Carotid Surgery Trial; NASCET, North American Symptomatic Carotid Endarterectomy Trial.

**Figure 8 diagnostics-13-02356-f008:**
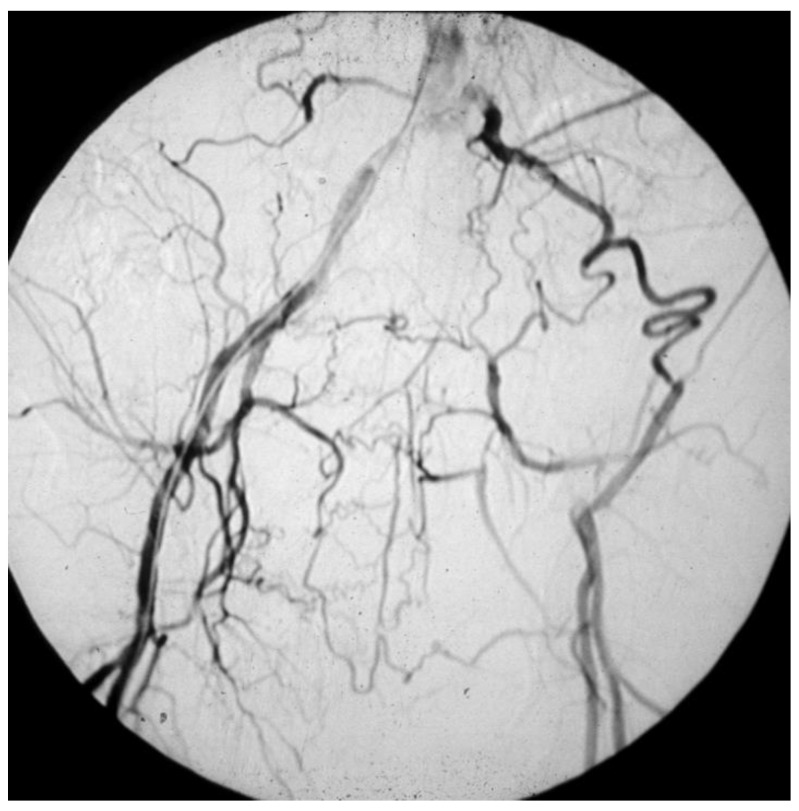
Arteriography showing obstruction of the left iliac axis, with an efficient collateral pathway revascularizing the common femoral artery. Depending on his/her age and lifestyle, this patient can eventually be asymptomatic or paucisymptomatic for intermittent claudication. An accurate physical examination (detecting the absence of the left femoral pulse), and the subsequent left ankle-brachial index (founded to be reduced) can easily allow for this patient to be classified as vasculopathic.

**Figure 9 diagnostics-13-02356-f009:**
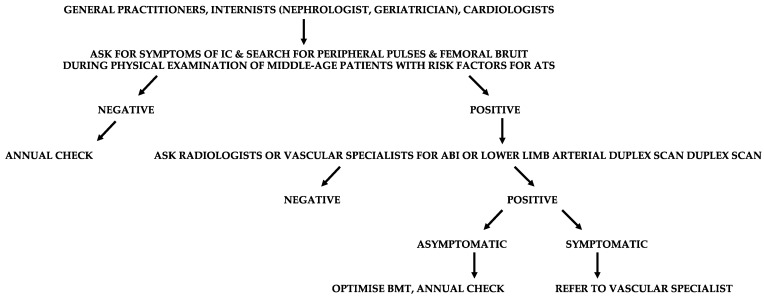
Suggested flow-chart for the detection of patients with lower extremity arterial disease (LEAD). IC, intermittent claudication; ATS, atherosclerosis; ABI, ankle-brachial index; BMT, best medical therapy.

**Figure 10 diagnostics-13-02356-f010:**
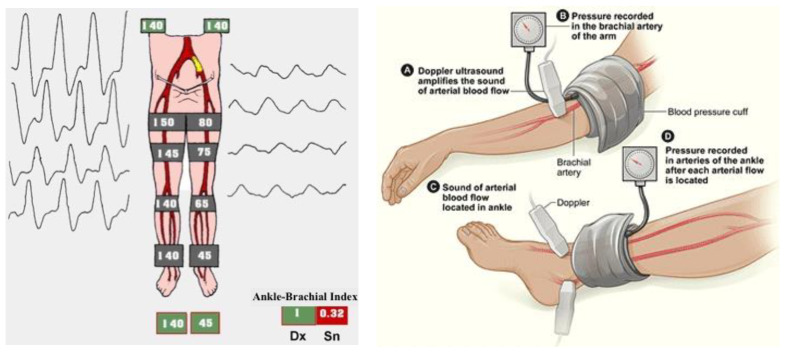
On the left, the ankle-brachial index (ABI), an absolute number which inversely correlates with the severity of lower extremity arterial disease (from https://www.aemmedi.it/files/la_scuola_AMD/2014/diabete_e_arteriopatia/9a_SimoniIparte.pdf (accessed on 1 May 2023)). On the right, the ABI’s measurement technique (from Wikimedia: https://www.nhlbi.nih.gov/health/health-topics/topics/pad/diagnosis (accessed on 1 May 2023)).

**Figure 11 diagnostics-13-02356-f011:**
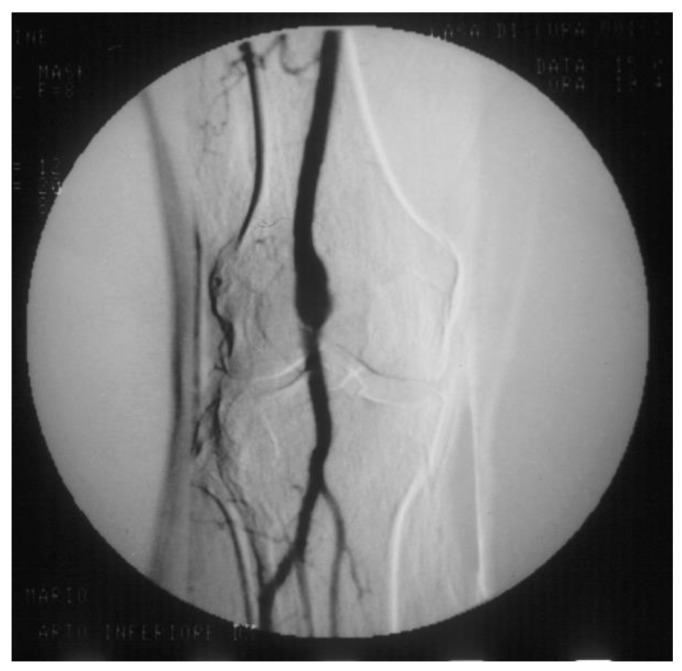
Arteriography showing popliteal artery aneurysm. The physiologic flexion movement of the knee can dislocate part of the mural thrombus, which embolizes and occludes some tibial arteries, giving rise to clinical pictures ranging from an asymptomatic state to intermittent claudication, or chronic limb-threatening ischemia, or acute limb ischemia.

**Figure 12 diagnostics-13-02356-f012:**
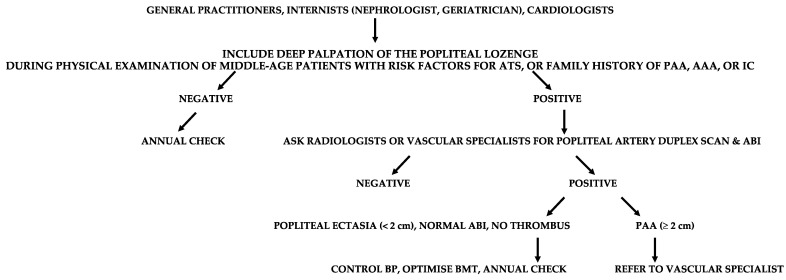
Suggested flow-chart for the detection of patients with popliteal artery aneurysm (PAA). ATS, atherosclerosis; AAA, abdominal aortic aneurysm; IC, intermittent claudication; ABI, ankle-brachial index; BP, blood pressure.

**Figure 13 diagnostics-13-02356-f013:**
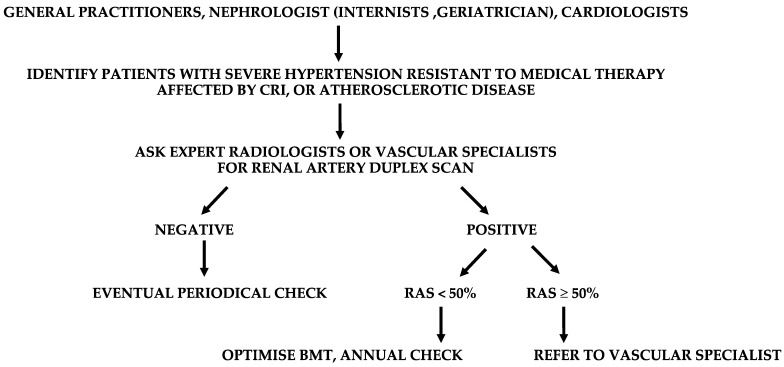
Suggested flow-chart for the detection of patients with renal artery stenosis (RAS). CRI, chronic renal insufficiency; BMT, best medical therapy.

**Figure 14 diagnostics-13-02356-f014:**
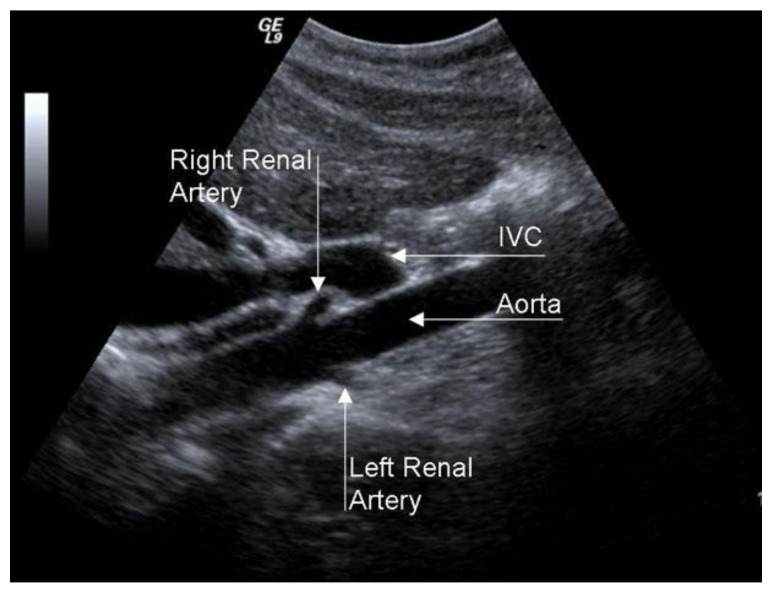
Duplex scan of the renal arteries (long axis view). IVC, inferior vena cava.

**Figure 15 diagnostics-13-02356-f015:**
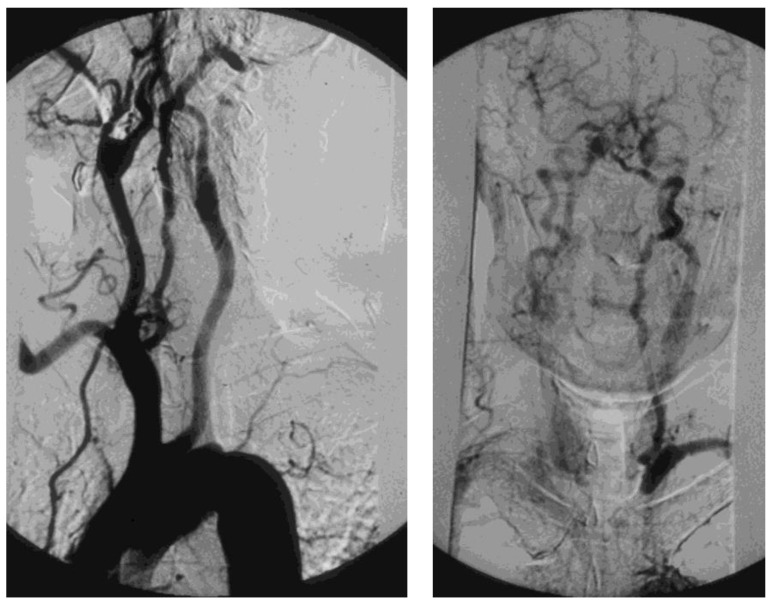
Arteriography showing occlusion of the left prevertebral subclavian artery at its origin (on the **left**): the late angiogram (on the **right**) demonstrates the blood flow inversion in the left vertebral artery, revascularizing the left subclavian artery after the occlusion, that is subclavian steal syndrome.

**Figure 16 diagnostics-13-02356-f016:**
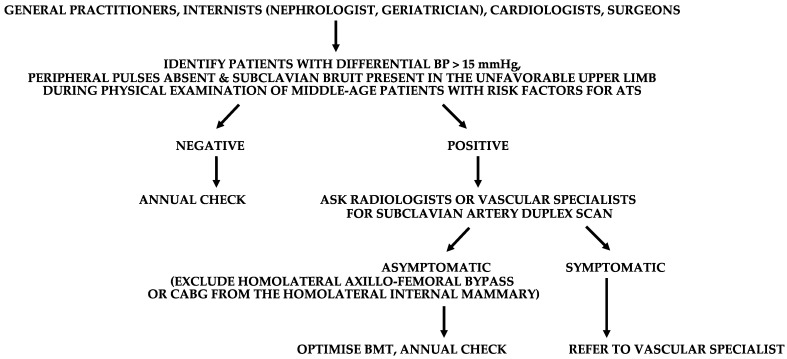
Suggested flow-chart for the detection of patients with prevertebral subclavian artery stenosis (PSAS). BP, blood pressure. ATS, atherosclerosis. BMT, best medical therapy. CABG, coronary artery bypass graft.

## Data Availability

Data sharing not applicable.

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
