# Peer review of "Focus on the Most Common Paucisymptomatic Vasculopathic Population, from Diagnosis to Secondary Prevention of Complications"

_diagnostics, 2023, doi:10.3390/diagnostics13142356_

Round 1

Reviewer 1 Report

This is an interesting review that aimed to evaluate the most common PADs and aneurysmal diseases, in the attempt to summarize some key points which should selectively drive screen

The abstract and manuscript are well-written, and the information appears complete. The background information is somewhat limited but there are enough studies referenced to support the paper.

The methods are described in good detail.

I consider that the authors should systematize the information in the tables much better so that the data presented is easier to follow.

Minor editing of the English language are required.

Author Response

We sincerely thank you for the appreciation of our manuscript. Following are our answer to your comment.

Q: “This is an interesting review that aimed to evaluate the most common PADs and aneurysmal diseases, in the attempt to summarize some key points which should selectively drive screen 

The abstract and manuscript are well-written, and the information appears complete. The background information is somewhat limited but there are enough studies referenced to support the paper.

The methods are described in good detail.

I consider that the authors should systematize the information in the tables much better so that the data presented is easier to follow.”

A: We have added flowcharts to the manuscript to summarize the information, whenever possible.

Reviewer 2 Report

Thank you for the opportunity to revise your work. You tried to provide an overview of aneurysmal to LEAD diagnosis. As you understand to cover this field adequately a few more pages are needed. This drives to a manuscript which does not really extend current knowledge. Probably you should focus on one field-vascular bed. In addition, a few sentences are missing references while reported with a quite strong language. 

What has driven these patients to a doctor? Do you report only on CAD cases? If so, clarify it in text. It is important to define the study population.

And if so, which doctor evaluates the patient? A general practitioner, a cardiologist or a vascular specialist? Who sets the indication for further evaluation with DUS each time? 

In addition, it would be very helpful to define a scheme for each pathology and try to follow it in text conduction. Some parts seem confusing or should be placed elsewhere in text. In addition, it is very difficult to cover the whole spectrum of each pathology; etiology, clinical symptoms and signs,  diagnostic tools, imaging. A structural reconstruction is mandatory for this article. 

No significant linguistic mistakes exist.

Author Response

We sincerely thank you for the effort and suggestions to improve our manuscript. Following are our answers to your comments.

Q: “You tried to provide an overview of aneurysmal to LEAD diagnosis. As you understand to cover this field adequately a few more pages are needed. This drives to a manuscript which does not really extend current knowledge. Probably you should focus on one field-vascular bed. In addition, a few sentences are missing references while reported with a quite strong language.” 

A: We added a few more pages to the manuscript (from the original 12 to the actual 22 pages), and the missing references.

Indeed, our goal was not to extend current knowledge (Diagnostics is a journal read by a large medical community, not specifically by vascular surgeons), while to raise awareness among the medical community to the evergreen central role covered by an accurate clinical history and a deep physical examination, which remain key factors to then select those individuals who should undergo first level instrumental investigations to confirm or exclude the clinical suspect of vasculopathy. This is in accordance with one of Diagnostics’ scopes, that is medical screening. We have better defined the target of our manuscript adding a sentence in the first paragraph.

Q: “What has driven these patients to a doctor? Do you report only on CAD cases? If so, clarify it in text. It is important to define the study population.

And if so, which doctor evaluates the patient? A general practitioner, a cardiologist or a vascular specialist? Who sets the indication for further evaluation with DUS each time?” 

A: We have specified the clinical journey of the patients, and the professionals most involved in each disease.

Q; “In addition, it would be very helpful to define a scheme for each pathology and try to follow it in text conduction. Some parts seem confusing or should be placed elsewhere in text. In addition, it is very difficult to cover the whole spectrum of each pathology; etiology, clinical symptoms and signs,  diagnostic tools, imaging. A structural reconstruction is mandatory for this article.” 

A: as stated in the first paragraph of the manuscript, our intention was “for each vascular disease described in this article we aim to answer two questions: 1) what and to whom screening for, and 2) how to prevent cardiovascular complications”. We tried to follow this order, that is to answer the points 1) and 2) in each vascular disease faced. However, we agree with you that our logical sequence may not be always intuitive. So we have amended the text, with the aim of better explaining the description of each of the two points faced in each vascular disease.

Furthermore, we did not expect to cover each vascular disease faced in an all encompassing fashion, i.e. dealing with a complete review of its etiology, clinical symptoms and signs, diagnostic tools, and imaging. However, we agree with your suggestion that these aspects should have been explored more in depth. So, we have implemented these points but trying not to deviate from the main two purposes of the manuscript, that is to answer the points 1) and 2) as declared in the Introduction section.

The manuscript has been fully checked for English grammar by a professional translator.

Round 2

Reviewer 2 Report

Dear Authors please find attached my comments. Your work is quite improved but further modifications are needed. 

We cannot screen anyone for PAD. It is not cost-effective and realistic. Could we imagine if a patient visiting a doctor for any other age to be controlled for any available disease? For example, lupus has fatal complications when diagnosed late. Should we check the entire population for lupus? I think you should reorganize the introduction focusing on high-risk populations that may benefit for further investigation.

Give definition of what it is considered PAD.

Please think replacing “screening” with “primary prevention”.

“It primarily affects male patient, and often manifests with abdominal or back pain and hypotension: too late to avoid an overall mortality rate that can be up to 81% [3,4].” Women are at higher risk for rupture when presenting with AAA. Please rephrase. AAA is more common in males but if present in females presents a higher rupture risk. As you state later in text.

“Sometimes, an intact asymptomatic AAA is accidentally detected during instrumental investigations conducted in the context of other disease: for instance, duplex scan of the liver or of the prostate, or computed tomography (CT-scan) or magnetic resonance imaging (MRI) of the column. Indeed, these are among the luckiest vasculopathic patients.” Your last sentence is a bit “enthusiastic”. Please eliminate.

In page 5 you probably tried to provide a scheme. But it is not readable. Please consider recreating as an image.

“Chronic obstructive pulmonary disease and active smoking are risk factors associated with AAA development.” Probably you would like to say “expansion” as this paragraph refers to growth rates after diagnosis.

“Atrial fibrillation is another major cause of cerebral embolization, causing more transient ischemic attacks (TIAs) than strokes; these latter are less sudden, disabling, and lethal than strokes of carotid origin [28].” This phrase should be eliminated or transferred in the beginning of this chapter.

“Analyzing in detail, EICS prevalence related to age and sex, a large USA and UK population screening of 2.5 million people revealed that the prevalence of EICS is 2% and 2.3% between ages 60-69, 3.6% and 6% for age 70-79, and 5% and 7.5% when >80 years old, for women and men, respectively.” Please rephrase to “A large USA and UK population EICS screening of 2.5 million people EICS showed that prevalence of EICS is 2% and 2.3% between ages 60-69, 3.6% and 6% for age 70-79, and 5% and 7.5% when >80 years old, for women and men, respectively.

“DSA is the gold standard to determine the extent of EICS,” The ESVS guidelines suggest DSA only in controversial findings. DUS operated by two different operators or DUS plus CTA or MRA is enough. Please consider rephrasing.

“Chronic renal insufficiency (CRI) is another risk factor for stroke in patients with asymptomatic EICS [46].” This phrase should be in “ How to prevent an ischemic stroke of EICS origin”

“Fortunately, if compared to the vascular diseases dealt with before, LEAD is more symptomatic, hence facilitating its diagnosis. However, only one-third of LEAD patients present the typical intermittent claudication (IC), a cramping pain in some muscles of the lower limbs,..” Add references.

Needs revision throughout the text. A few sentences are a bit difficult to understand. 

Author Response

Answers to reviewer #2:

Once again, we sincerely thank you for the further efforts and suggestions to improve our manuscript. Indeed, your comments are very useful. Following are our answers and corrections.

Q: “We cannot screen anyone for PAD. It is not cost-effective and realistic. Could we imagine if a patient visiting a doctor for any other age to be controlled for any available disease? For example, lupus has fatal complications when diagnosed late. Should we check the entire population for lupus? I think you should reorganize the introduction focusing on high-risk populations that may benefit for further investigation.

Give definition of what it is considered PAD.” 

A: Our focus has been restricted to “…middle-age patients with one or more risk factors for atherosclerosis go to medical consultation, for instance to a cardiologist or a pneumoligist”.

PAD has been replaced with “arterial diseases”.

Q: Please think replacing “screening” with “primary prevention”, 

A: Actually, we prefer to maintain the term “screening”, without changing to “primary prevention”. For instance, primary prevention of carotid stenosis would signify better control and treatment of atherosclerosis’ risk factors, which is not the real focus of our manuscript.

Q: “It primarily affects male patient, and often manifests with abdominal or back pain and hypotension: too late to avoid an overall mortality rate that can be up to 81% [3,4].” Women are at higher risk for rupture when presenting with AAA. Please rephrase. AAA is more common in males but if present in females presents a higher rupture risk. As you state later in text.” 

A: Rephrased.

Q: ““Sometimes, an intact asymptomatic AAA is accidentally detected during instrumental investigations conducted in the context of other disease: for instance, duplex scan of the liver or of the prostate, or computed tomography (CT-scan) or magnetic resonance imaging (MRI) of the column. Indeed, these are among the luckiest vasculopathic patients.” Your last sentence is a bit “enthusiastic”. Please eliminate.”

A: Eliminated.

Q: “In page 5 you probably tried to provide a scheme. But it is not readable. Please consider

recreating as an image.”

A: Figure 4 at page 5, as well as all the other flow-charts, have been recreated as images.

Q: “Chronic obstructive pulmonary disease and active smoking are risk factors associated with AAA development.” Probably you would like to say “expansion” as this paragraph refers to growth rates after diagnosis.

A: Corrected.

Q: “Atrial fibrillation is another major cause of cerebral embolization, causing more transient ischemic attacks (TIAs) than strokes; these latter are less sudden, disabling, and lethal than strokes of carotid origin [28].” This phrase should be eliminated or transferred in the beginning of this chapter.

A: The first half of this sentence has been moved to the beginning of this chapter, while the second half has been eliminated.

Q: “Analyzing in detail, EICS prevalence related to age and sex, a large USA and UK population screening of 2.5 million people revealed that the prevalence of EICS is 2% and 2.3% between ages 60-69, 3.6% and 6% for age 70-79, and 5% and 7.5% when >80 years old, for women and men, respectively.” Please rephrase to “A large USA and UK population EICS screening of 2.5 million people EICS showed that prevalence of EICS is 2% and 2.3% between ages 60-69, 3.6% and 6% for age 70-79, and 5% and 7.5% when >80 years old, for women and men, respectively.

A: Rephrased.

Q: “DSA is the gold standard to determine the extent of EICS,” The ESVS guidelines suggest DSA only in controversial findings. DUS operated by two different operators or DUS plus CTA or MRA is enough. Please consider rephrasing.

A: The phrase has been eliminated, to avoid going in to too much specialistic detail for the intended readers of this manuscript.

Q: “Chronic renal insufficiency (CRI) is another risk factor for stroke in patients with asymptomatic EICS [46].” This phrase should be in “ How to prevent an ischemic stroke of EICS origin”

A: The sentence has been moved up to at the beginning of the section.

Q: “Fortunately, if compared to the vascular diseases dealt with before, LEAD is more symptomatic, hence facilitating its diagnosis. However, only one-third of LEAD patients present the typical intermittent claudication (IC), a cramping pain in some muscles of the lower limbs,..” Add references.

A: Reference added.

Q: Comments on the Quality of English Language. Needs revision throughout the text. A few sentences are a bit difficult to understand. 

A: The manuscript has been, once again, fully checked for English grammar by a professional translator.

Round 3

Reviewer 2 Report

No further comment.